# Learning Bounds for Risk-sensitive Learning

**Jaeho Lee**[†*]    **Sejun Park**[‡]    **Jinwoo Shin**[‡†]
Korea Advanced Institute of Science and Technology (KAIST)
[†] School of Electrical Engineering,   [‡] Graduate School of AI

## Abstract

In risk-sensitive learning, one aims to find a hypothesis that minimizes a risk-averse (or risk-seeking) measure of loss, instead of the standard expected loss. In this paper, we propose to study the generalization properties of risk-sensitive learning schemes whose optimand is described via optimized certainty equivalents (OCE): our general scheme can handle various known risks, e.g., the entropic risk, mean-variance, and conditional value-at-risk, as special cases. We provide two learning bounds on the performance of empirical OCE minimizer. The first result gives an OCE guarantee based on the Rademacher average of the hypothesis space, which generalizes and improves existing results on the expected loss and the conditional value-at-risk. The second result, based on a novel variance-based characterization of OCE, gives an expected loss guarantee with a suppressed dependence on the smoothness of the selected OCE. Finally, we demonstrate the practical implications of the proposed bounds via exploratory experiments on neural networks.

## 1   Introduction

The systematic minimization of the quantifiable uncertainty, or *risk* [28], is one of the core objectives in all disciplines involving decision-making, e.g., economics and finance. Within machine learning contexts, strategies for *risk-aversion* have been most actively studied under sequential decision-making and reinforcement learning frameworks [25, 9], giving birth to a number of algorithms based on Markov decision processes (MDPs) and multi-armed bandits. In those works, various risk-averse measures of loss have been used as a minimization objective, instead of the risk-neutral expected loss; popular risk measures include entropic risk [25, 7, 8], mean-variance [47, 17, 35], and a slightly more modern alternative known as conditional value-at-risk (CVaR [19, 11, 50]). Yet, with growing interest to the societal impacts of machine intelligence, the importance of risk-aversion under *non-sequential* scenarios has also been spotlighted recently. For instance, Williamson and Menon [53] give an axiomatic characterization of the fairness risk measures, and propose a convex fairness-aware objective based on CVaR. Also, Curi et al. [14] empirically demonstrate the effectiveness of their CVaR minimization algorithm to account for the covariate shift in the data-generating distribution.

The advantage of *risk-sensitive* (either *risk-seeking* or *risk-averse*) objectives in machine learning, however, is not limited to tasks involving social considerations. Indeed, there exists a rich body of works which *implicitly* propose to minimize risk-sensitive measures of loss, as a technique to better optimize the standard expected loss. For example, the idea of prioritizing low-loss samples for learning is prevalent in noisy label handling [22] or curriculum learning [30]. In those contexts, high-loss samples are viewed as either mislabeled, or correctly labeled but detrimental to training dynamics due to their "difficulty." Such algorithms can be viewed as implicitly optimizing a risk-seeking counterpart of CVaR (see Section 2.2). Contrarily (and ironically), it is also common to focus on high-loss samples to improve the model accuracy or accelerate the optimization [10]. Such algorithms can be viewed as minimizing risk-averse measures of loss; for instance, learning with *average top-$k$ loss* [16] is equivalent to the CVaR minimization when the number of samples is fixed.

---

[*]To be corresponded with: `jaeho-lee@kaist.ac.kr`

Given this widespread use of risk-sensitive learning algorithms, theoretical understandings of their generalization properties are still limited. For risk-seeking learning, the risk measure being minimized is typically not explicitly stated; see [22], for instance. For risk-averse learning, existing theoretical results are focused on validating the stability and convergence of the algorithm (e.g. [35]), instead of providing generalization/excess-risk guarantees. Some exceptions in this respect are the recent works on CVaR [16, 14, 48]; the guarantees, however, are highly specialized for the algorithmic setups considered, such as support vector machines with reproducing kernel Hilbert spaces [16], finite function class[2] [14], or requiring additional smoothness assumptions [48].

To fill this gap, we propose to study risk-sensitive learning schemes under a statistical learning theory viewpoint [23], where the focus is on the convergence properties of the risk measure itself; learning algorithms are simply abstracted as a procedure of finding a hypothesis minimizing the target risk measure on training data. To discuss various risk-sensitive measures under a unified framework, we rejuvenate the notion of optimized certainty equivalent (OCE [5]). With a careful choice of the *disutility* function governing the deviation penalty, OCE covers a wide range of risk-averse measures including the entropic risk, mean-variance, and CVaR (see Section 2). To formalize risk-seeking learning schemes, we newly define *inverted OCE* as a natural counterpart of OCE; inverted OCE covers learning algorithms that only utilize a fraction of samples with smallest losses.

Under this general framework, we establish two performance guarantees for the empirical OCE minimization (EOM) procedure (see Section 3); we also provide analogous results for inverted OCEs.

- Theorem 4 provides a general bound on the excess OCE of the EOM hypothesis via Rademacher averages. For the case of CVaR, the bound provides a first data-dependent bound that improves or recovers the existing data-independent bounds (e.g., VC dimension). For the case of expected loss, the bound recovers the standard risk guarantee. The proof is based on the contraction properties [31] of a product space constructed with the original hypothesis space and dual parameter space.
- Theorem 7 controls the expected loss of the EOM hypothesis via a novel variance-based characterization of OCE (Lemma 6). In contrast to the OCE guarantee in Theorem 4, the expected loss guarantee does not depend crucially on the properties of the target OCE measure in the *realizable case*, i.e., the hypothesis space is rich enough to contain a hypothesis with an arbitrarily small loss.

Finally, we empirically validate an implication of Lemma 6 that EOM can be relaxed to the sample variance penalization (SVP) procedure. The relaxed version is known to enjoy stronger generalization properties, making the algorithm an attractive candidate to be considered as an *alternative baseline method* for the OCE minimization. In our experiments on CIFAR-10 [29] with ResNet18 [24], we find that batch-based SVP indeed outperforms batch-based CVaR minimization (see Section 4).

All proofs are deferred to the Appendix A.

**Notations.** For a real number $t \in \mathbb{R}$, we let $[t]_+ := \max\{0, t\}, [t]_- := \max\{0, -t\}$. We write $\mathbb{R}_+$ to denote the set of all nonnegative real numbers. When $t \in [0, 1]$, we let $\bar{t} := 1 - t$. For a real-valued function $\phi : \mathbb{R} \to \mathbb{R}$, we write $\partial \phi(\cdot)$ to denote its subgradient set, and $\text{Lip}(\phi)$ to denote its Lipschitz constant (on the considered domain). The *pushforward* measure of a distribution $P$ by a mapping $f$, i.e., the distribution of $f(Z)$ where $Z \sim P$, is denoted by $f_\sharp P$. For the probability distribution $Q$ of a real random variable, the notation $\mathfrak{q}(\alpha; Q)$ denotes the quantile function $\inf\{t \in \mathbb{R} \mid \alpha \leq F_Q(t)\}$, where $F_Q$ is the cumulative distribution function of $Q$. All logs are base $e$.

## 2  Measures for risk-sensitive learning

We start from the standard statistical learning framework [23]. We have a class $\mathcal{P}$ of probability measures called *data-generating distributions*, defined on a measurable *instance space* $\mathcal{Z}$. We are also given a *hypothesis space* $\mathcal{F}$ of measurable functions $f : \mathcal{Z} \to \mathbb{R}_+$, quantifying the loss incurred by a decision rule when applied to a data instance $z \in \mathcal{Z}$. A standard measure to aggregate samplewise losses of a hypothesis over a population of data instances is to take an *expected loss*,[3] defined as

$$R(f) := \mathbf{E}_P[f(Z)] = \int_{\mathcal{Z}} f(z) P(\mathrm{d}z). \tag{1}$$

Table 1: Popular OCE risks in machine learning literature, and corresponding disutility functions.

| Name | Definition | Disutility function |
|---|---|---|
| Expected loss | $\mathbf{E}[f(Z)]$ | $\phi(t) = t$ |
| Entropic risk | $\frac{1}{\gamma} \log \mathbf{E}[e^{\gamma f(Z)}]$ | $\phi_\gamma(t) = \frac{1}{\gamma} e^{\gamma t} - \frac{1}{\gamma}$ |
| Mean-variance | $\mathbf{E}[f(Z)] + c \cdot \mathbf{E}[(f(Z) - \mathbf{E}[f(Z)])^2]$ | $\phi_c(t) = t + ct^2$ |
| Conditional Value-at-Risk[†] | $\mathbf{E}[f(Z) \mid f(Z) > \mathsf{q}(1-\alpha; f_\sharp P)]$ | $\phi_\alpha(t) = \frac{1}{\alpha}[t]_+$ |

[†] The conditional expectation representation holds only for the $(f, P)$ pairs generating continuous pushforwards $f_\sharp P$. A more general definition that covers the discrete case can be found in [41].

We assume that the data-generating distribution $P \in \mathcal{P}$ is not known to the learner. Instead, the learner is assumed to have an access to $n$ copies of training samples $Z^n = (Z_1, \ldots, Z_n)$ independently drawn from $P$. Then, the expected loss can be estimated by the *empirical loss*

$$R_n(f) := \mathbf{E}_{P_n}[f(Z)] = \frac{1}{n} \sum_{i=1}^n f(Z_i), \tag{2}$$

where $P_n$ denotes the empirical distribution of training samples. Both expected loss and empirical loss are *risk-neutral* measures that assign uniform weight on the samples regardless of their losses.

## 2.1 Risk-averse measures: optimized certainty equivalents

Among the diverse set of measures for risk-aversion in economics (see Appendix C for details), we focus on the *optimized certainty equivalents* (OCE) introduced by Ben-Tal and Teboulle [5].

**Definition 1** (OCE risk). *Let the disutility function $\phi : \mathbb{R} \to \mathbb{R} \cup \{+\infty\}$ be a nondecreasing, closed, convex function with $\phi(0) = 0$ and $1 \in \partial\phi(0)$. Then, the corresponding OCE risk is given as*[4]

$$\mathsf{oce}^\phi(f; P) := \inf_{\lambda \in \mathbb{R}} \left\{ \lambda + \mathbf{E}_P[\phi(f(Z) - \lambda)] \right\}. \tag{3}$$

Definition 1, having its root in the expected utility theory [52], may look mysterious at first glance. To demystify a little bit, consider the following reparametrization: define the *excess disutility* as the difference of the disutility and the identity $\varphi(t) := \phi(t) - t$. From Definition 1, we know that the excess disutility $\varphi$ is a nonnegative, convex function satisfying $\varphi(0) = 0$, with a nondecreasing $\varphi(t) + t$. Then, the OCE risk can be written as

$$\mathsf{oce}(f) = \inf_{\lambda \in \mathbb{R}} \left\{ \lambda + \mathbf{E}_P[\varphi(f(Z) - \lambda) + f(Z) - \lambda] \right\} = R(f) + \inf_{\lambda \in \mathbb{R}} \mathbf{E}_P[\varphi(f(Z) - \lambda)]. \tag{4}$$

In other words, OCE additionally penalizes the expected deviation of the random object $f(Z)$ from some optimized anchor point $\lambda$. The penalty is described by the selection of a "bowl-shaped" excess disutility $\varphi$ (or equivalently, the selection of disutility $\phi$).

With a careful choice of $\phi$, Definition 1 covers a wide range of risk-averse measures used in machine learning literature, including the expected loss, entropic risk, mean-variance, and CVaR; popular OCE risks and corresponding choices of disutility are summarized in Table 1. The measures have been used in the following machine learning contexts. (1) Entropic risk: The risk has been used in one of the earliest works on risk-sensitive MDPs [25], and is often revisited in modern reinforcement learning contexts [7, 8, 39]. In a concurrent work, Li et al. [33] re-introduces the entropic risk to enhance outlier-robustness and fairness. (2) Mean-variance: Markowitz's mean-variance analysis [36] is typically relaxed to the variance regularization in the context of MDPs [17, 35], multi-armed bandits [43, 51], and reinforcement learning [47, 1]. (3) CVaR: CVaR is used in more recent works on risk-averse reinforcement learning regarding bandits [19, 9] and MDPs [11, 50]. CVaR also enjoys connections to distributional robustness and fairness under general learning scenarios [53, 14, 48].

Similar to the expected loss, the OCE risk of a data-generating distribution can be estimated from the samples by using the empirical distribution as a proxy measure: we define the *empirical OCE risk* as

$$\mathsf{oce}_n^\phi(f) := \mathsf{oce}^\phi(f; P_n) = \inf_{\lambda \in \mathbb{R}} \left\{ \lambda + \frac{1}{n} \sum_{i=1}^n \phi(f(Z_i) - \lambda) \right\}. \tag{5}$$

The empirical OCE *underestimates* the population OCE in general, due to its variational definition. Indeed, the OCE risk for a mixture distribution $\alpha P + \bar{\alpha} Q$ is always greater than or equal to the weighted average of OCE risks $\alpha \mathsf{oce}(f; P) + \bar{\alpha} \mathsf{oce}(f; Q)$, as the inequality $\inf_\lambda \left\{ g_1(\lambda) + g_2(\lambda) \right\} \geq \inf_\lambda g_1(\lambda) + \inf_\lambda g_2(\lambda)$ holds. From this observation, one may expect a slower two-sided uniform convergence of empirical OCE than empirical loss; this intuition is confirmed later (see Lemma 3).

We also note that OCE risks satisty the following properties, which enable an efficient computation and optimization (see [6] for derivations): (a) Convexity, i.e., $\mathsf{oce}(\alpha f_1 + \bar{\alpha} f_2) \leq \alpha \mathsf{oce}(f_1) + \bar{\alpha} \mathsf{oce}(f_2)$, (b) Shift-additivity, i.e., $\mathsf{oce}(f + c) = \mathsf{oce}(f) + c$, (c) Monotonicity, i.e., if $f_1(Z) \leq f_2(Z)$ with probability 1, then $\mathsf{oce}(f_1) \leq \mathsf{oce}(f_2)$. Convexity is especially useful whenever the loss function underlying the hypotheses are also convex; interested readers are referred to Appendix C.

## 2.2 Risk-seeking measures: inverted OCEs

Unlike in financial economics literature, it often occurs in machine learning schemes [30, 22] to focus on the minimization of losses on *easy examples* (i.e., the samples already with low loss) and disregard hard examples. To formally address such learning algorithms, we propose considering the following family of risk-seeking measures constructed by *inverting* OCE risks.

**Definition 2** (Inverted OCE risk)**.** *Let $\phi : \mathbb{R} \to \mathbb{R} \cup \{+\infty\}$ be a nondecreasing, closed, convex function with $\phi(0) = 0$ and $1 \in \partial\phi(0)$. Then, the corresponding inverted OCE risk is given as*

$$\overline{\mathsf{oce}}^\phi(f; P) := \sup_{\lambda \in \mathbb{R}} \left\{ \lambda - \mathbf{E}_P[\phi(\lambda - f(Z))] \right\}. \tag{6}$$

We call the measure (6) an "inverted" OCE risk due to the following reason: Roughly speaking, the inverted OCE risk is designed to treat the sample at bottom-$\alpha$ loss quantile as the OCE risk treats the sample at top-$\alpha$ loss quantile. This goal can be achieved by defining the inverted OCE risk to satisfy $\overline{\mathsf{oce}}(f) = -\mathsf{oce}(-f)$, which gives the form (6). Analogously to Eq. (4), the inverted OCE risk can be written alternatively as

$$\overline{\mathsf{oce}}(f) = R(f) - \inf_{\lambda \in \mathbb{R}} \mathbf{E}_P[\varphi(\lambda - f(Z))], \tag{7}$$

where again $\varphi(t) = \phi(t) - t$. In other words, the inverted OCE risk rewards the deviation from the optimized anchor $\lambda$, using the excess utility function $-\varphi(-t)$ to shape the reward.

It is straightforward to see that inverted OCE risks can be used to describe the algorithms that disregard samples with high loss. For example, Han et al. [22] propose the following algorithm to handle *noisy labels*: two models are trained simultaneously, by selecting and feeding $\alpha$-fraction of samples with the lowest loss to each other. Such a training objective can be described as an inverted version of CVaR, i.e., by using $\phi(t) = \frac{1}{\alpha}[t]_+$. Indeed, we can see the equivalence from the following proposition (see Appendix A.1 for the proof).

**Proposition 1** (Average bottom-$k$ loss as inverted CVaR)**.** *Let $k \in \mathbb{N}, k \leq n$ be the desired number of samples. Then, by the choice of disutility function $\phi(t) = \frac{n}{k}[t]_+$, i.e., $\alpha = \frac{k}{n}$, we get*

$$\overline{\mathsf{oce}}^\phi(f; P_n) = \frac{1}{k} \sum_{i=1}^{k} f(Z_{\pi(i)}), \tag{8}$$

*where $\pi(i)$ denotes the index of the sample with $i$-th smallest value of $f(\cdot)$ among $Z^n$.*

The proposed notion of inverted OCE can thus be viewed as a generalized class of optimands for *easy example first* algorithms, that comes with theoretical performance guarantees (Theorems 4 and 7). We note that this class also includes a "softer" variant of the algorithm considered in Proposition 1, where a weighted sum of sample losses is taken with weights $\left\{ \frac{\gamma_1}{n}, \frac{\gamma_2}{n} \right\}$ instead of $\left\{ \frac{1}{n\alpha}, 0 \right\}$ for the bottom-$\alpha$ fractions and top-$\bar{\alpha}$ fraction, respectively; we naturally assume that $0 \leq \gamma_2 < 1 < \gamma_1$ and $\alpha = \frac{1-\gamma_2}{\gamma_1-\gamma_2}$ holds. Indeed, one can simply choose $\phi(t) = \gamma_1[t]_+ - \gamma_2[t]_-$ to get the desired risk.

Given this connection, can we explain the empirical robustness of the noisy label handling algorithms (such as [22]) by analyzing the properties of inverted OCE risks? While this question is not under the main scope of this paper, we provide a partial answer to this question by analyzing the *influence function* [21], which is one of the key notions in the discipline of robust statistics. The function

measures the sensitivity of a statistic to a distributional shift which may represent an outlier or contaminated data. Formally, the influence function of a statistic $\rho : \mathcal{P} \to \mathbb{R}_+$ with respect to $z^* \in \mathcal{Z}$ is given as

$$\mathsf{IF}(z^*; P, \rho) := \lim_{\varepsilon \to 0^+} \frac{\rho(\bar{\varepsilon}P + \varepsilon \Delta_{z^*}) - \rho(P)}{\varepsilon}, \tag{9}$$

where $\Delta_{z^*}$ denotes the point probability mass at $z^*$, and $P$ denotes the distribution of uncontaminated samples. If we use the OCE risk as a target statistic (i.e., $\rho(\cdot) = \mathsf{oce}^\phi(f; \cdot)$), then the influence function can be viewed as a sensitivity of OCE minimization procedure against a distributional contamination. As a historical remark, we note that the influence function (9) is typically studied under a parametric framework, e.g., gauging the robustness of an estimator of distributional parameters such as moments (see the seminal treatise of Huber and Ronchetti [26] for a comprehensive overview). Nevertheless, we are not the first to analyze the influence function under a nonparametric scenario; for instance, Christmann and Steinwart [12] have studied the influence function of penalized empirical risk minimization procedure.

In the following proposition, we show that the inverted versions of popular OCE measures have better robustness characteristics than the expected loss (see Appendix A.2 for the proof).

**Proposition 2** (Influence function of $\overline{\mathsf{oce}}$). *The influence function for the inverted entropic risk and the inverted mean-variance are given as follows.*

- *Entropic risk:* $\frac{1}{\gamma} - \frac{1}{\gamma} \frac{e^{-\gamma f(z^*)}}{\mathbf{E}_P[e^{-\gamma f(Z)}]}$.

- *Mean-variance:* $f(z^*) - R(f) + c \left[ \mathbf{E}_P[(f(Z) - R(f))^2] - (f(z^*) - R(f))^2 \right]$.

*Whenever $f_{\sharp P}$ has a continuous density, then the influence function of inverted CVaR is given as*

- *CVaR:* $\frac{1}{\alpha} \mathbf{E}_P[\mathfrak{q}(\alpha; f_{\sharp P}) - f(Z)]_+ - \frac{1}{\alpha}[\mathfrak{q}(\alpha; f_{\sharp P}) - f(z^*)]_+$.

From Proposition 2, we observe that the influence functions of the example inverted OCE risks have a smaller *worse-case* value than the influence function of the expected loss, which is $f(z^*) - R(f)$. In particular, whenever there exists some $z^*$ such that $R(f)$ is arbitrarily large, then the influence function of the expected loss can grow arbitrarily large as well. On the other hand, influence functions of the example inverted OCE risks are bounded from above by

$$\frac{1}{\gamma}, \quad \frac{3}{4c} + c \cdot \mathbf{E}_P[(f(Z) - R(f))^2], \quad \frac{1}{\alpha} \mathbf{E}_P[\mathfrak{q}(\alpha; f_{\sharp P}) - f(Z)]_+, \tag{10}$$

respectively for inverted entropic risk, mean-variance, and CVaR.

## 3 Performance guarantees for empirical OCE minimizers

We now consider an *empirical OCE minimization* (EOM) procedure, finding

$$\widehat{f}_{\mathsf{eom}} := \arg\min_{f \in \mathcal{F}} \mathsf{oce}_n(f), \tag{11}$$

instead of the ordinary empirical risk minimization (ERM), which aims to minimize the empirical loss. Existing learning algorithms that implement EOM, either explicitly or implicitly, can be roughly categorized into two categories, depending on their purposes. In the works of the first category (e.g. [16, 14, 48]), the primary goal is to minimize the population OCE risk (i.e., $\mathsf{oce}(f)$) for risk-aversion or fairness considerations. In the works of the second category (e.g. [10, 37, 33]), the ultimate goal is to optimize the population expected loss (i.e., $R(f)$), and risk-sensitive measures are used with the belief that minimizing the measures may help accelerate/stabilize the learning dynamics. To address both lines of research, we provide performance guarantees in terms of both OCE and expected loss. In particular, we show that the empirical OCE minimizer (11) has the OCE risk and the expected loss similar to those of

$$f^*_{\mathsf{oce}} := \arg\min_{f \in \mathcal{F}} \mathsf{oce}(f), \qquad f^*_{\mathsf{avg}} := \arg\min_{f \in \mathcal{F}} R(f), \tag{12}$$

(shown in Section 3.1 and Section 3.2, respectively). We also give analogous results for the empirical inverted OCE minimization (EIM), where the hypotheses achieving minimum empirical and population $\overline{\mathsf{oce}}$ will be denoted by $\widehat{f}_{\mathsf{eim}}$ and $f^*_{\overline{\mathsf{oce}}}$.

## 3.1 OCE guarantee via uniform convergence

First, we provide an excess OCE guarantee of the empirical OCE minimizer based on the uniform convergence of the empirical OCE risk to the population OCE risk. To formalize, recall that the *Rademacher average* [4] of a hypothesis space $\mathcal{F}$ given training samples $Z^n$ is defined as

$$\mathfrak{R}_n(\mathcal{F}(Z^n)) := \mathbf{E}_{\epsilon^n}\left[\sup_{f \in \mathcal{F}}\left\{\frac{1}{n}\sum_{i=1}^{n}\epsilon_i f(Z_i)\right\}\right], \tag{13}$$

where $\{\epsilon_i\}_{i=1}^{n}$ are independent Rademacher random variables, i.e., $\mathbf{P}[\epsilon_i = +1] = \mathbf{P}[\epsilon_i = -1] = \frac{1}{2}$. With this definition at hand, we can state our key lemma characterizing uniform convergence properties of OCE risks and inverted OCE risks (see Appendix A.3 for the proof).

**Lemma 3** (Uniform convergence). *Suppose that the hypothesis space is bounded, i.e. there exists some $M > 0$ such that $\sup_{z \in \mathcal{Z}} f(z) \leq M$ holds for all $f \in \mathcal{F}$. Then, for any $\delta \in (0, 1]$,*

$$\sup_{f \in \mathcal{F}}\left|\mathrm{oce}(f) - \mathrm{oce}_n(f)\right| \leq \mathrm{Lip}(\phi) \cdot \left(2\mathbf{E}[\mathfrak{R}_n(\mathcal{F}(Z^n))] + \frac{M(2 + \sqrt{\log(2/\delta)})}{\sqrt{n}}\right) \tag{14}$$

*holds with probability at least $1 - \delta$.*

*Moreover, the same bound holds whenever the $\mathrm{oce}, \mathrm{oce}_n$ are replaced by $\overline{\mathrm{oce}}, \overline{\mathrm{oce}}_n$.*

Similar to typical uniform convergence guarantees for the empirical loss [4], the bound (14) vanishes to zero at the rate $1/\sqrt{n}$ for standard hypothesis spaces whose expected Rademacher averages could be bounded from above by a $\mathcal{O}(1/\sqrt{n})$ term. Indeed, Lemma 3 closely recovers the usual uniform convergence bound for expected loss if we plug in $\phi(t) = t$, with a slack of $2M/\sqrt{n}$ that is small compared to the other terms. We also note that the Lipschitz constant of disutility functions cannot be smaller than one, and thus Lemma 3 cannot be used to guarantee a strictly faster convergence rate than the bound for the expected loss.

Lemma 3 generalizes and improves over existing guarantees on CVaR [44, 49, 14, 48]. Indeed, all previous results (up to our knowledge) are described in terms of data-independent complexity measures of the hypothesis space, e.g. VC-dimension; roughly, this is due to the proof technique relying on a direct use of union bound. In contrast, by considering a dual product space approach (see, e.g. [32]) combined with contraction principles [31], we arrive at the bound described via Rademacher averages. Rademacher average is a data-dependent complexity measure [4] which enjoys a significant benefit in the analysis of modern hypothesis spaces. Indeed, the data-dependency is considered an irreplaceable element to understanding the generalization properties of neural networks [54]. At the same time, Rademacher averages can be controlled by data-independent complexity measures such as VC-dimension, to recover existing results; see [4] for an extensive discussion.

Using Lemma 3, we give an excess OCE risk guarantee on the hypothesis minimizing the empirical OCE risk (see Appendix A.4 for the proof).

**Theorem 4** (OCE guarantee). *Suppose that the hypothesis space is bounded, i.e. there exists some $M > 0$ such that $\sup_{z \in \mathcal{Z}} f(z) \leq M$ holds for all $f \in \mathcal{F}$. Then, the empirical OCE minimizer (11) satisfies*

$$\mathrm{oce}(\widehat{f}_{\mathrm{eom}}) \leq \mathrm{oce}(f_{\mathrm{oce}}^*) + \mathrm{Lip}(\phi) \cdot \left(4\mathbf{E}[\mathfrak{R}_n(\mathcal{F}(Z^n))] + \frac{2M(2 + \sqrt{\log(2/\delta)})}{\sqrt{n}}\right), \tag{15}$$

*with probability at least $1 - \delta$. For the empirical inverted OCE minimizer, we analogously have*

$$\overline{\mathrm{oce}}(\widehat{f}_{\mathrm{eim}}) \leq \overline{\mathrm{oce}}(f_{\overline{\mathrm{oce}}}^*) + \mathrm{Lip}(\phi) \cdot \left(4\mathbf{E}[\mathfrak{R}_n(\mathcal{F}(Z^n))] + \frac{2M(2 + \sqrt{\log(2/\delta)})}{\sqrt{n}}\right), \tag{16}$$

*with probability at least $1 - \delta$.*

For sufficiently expressive hypothesis spaces, $\mathrm{oce}(f_{\mathrm{oce}}^*)$ will become close to zero, and the upper bound becomes directly proportional to the Lipschitz constant of the disutility function.

## 3.2 Expected loss guarantee via variance-based characterization

To establish expected loss guarantees for the empirical OCE minimizer, we give two inequalities relating moments of the loss population to the OCE risk. The first one follows directly from the definitions of oce and $\overline{\text{oce}}$ (see Appendix A.5 for the proof).

**Proposition 5** (Mean-based characterization). *For any $f, P$ and $\phi$, we have*

$$0 \le \overline{\text{oce}}(f) \le R(f) \le \text{oce}(f) \le \text{Lip}(\phi) \cdot R(f). \tag{17}$$

Combining Proposition 5 with Lemma 3, one can obtain an elementary expected loss guarantee on the EOM hypothesis: With probability at least $1 - \delta$, we have

$$R(\widehat{f}_{\text{eom}}) \le \text{Lip}(\phi) \cdot \left( R(f_{\text{avg}}^*) + 4\mathbf{E}[\mathfrak{R}_n(\mathcal{F}(Z^n))] + \frac{2M(2 + \sqrt{\log(2/\delta)})}{\sqrt{n}} \right). \tag{18}$$

The explanatory power of Ineq. (18), however, is clearly limited. To see this, consider a sufficiently expressive hypothesis space, so that one can always find a hypothesis perfectly fitting the training data. In this case, the EOM hypothesis also minimizes the expected loss, as we know that $R_n(f) \le \text{oce}_n(f)$ holds from Proposition 5. Then, one may expect an expected loss guarantee of the EOM hypothesis to be similar to that of the ERM hypothesis, not scaling with $\text{Lip}(\phi)$.

In light of this observation, we provide an alternative bound which relates OCE risks to both mean and variance of the loss population. For conciseness, we first introduce a shorthand notation for the *loss variance* of a hypothesis $f \in \mathcal{F}$.[5]

$$\sigma(f; P) := \sqrt{\mathbf{E}_P[(f(Z) - R(f))^2]}. \tag{19}$$

Now we can prove the following lemma bounding the difference of OCE risks and expected loss in terms of loss variance (see Appendix A.6 for the proof).

**Lemma 6** (Variance-based characterization). *Let $f : \mathcal{Z} \to \mathbb{R}_+$ be a bounded function, i.e. there exists some $M > 0$ such that $\sup_{z \in \mathcal{Z}} f(z) \le M$ holds. Then, we have*

$$C_\phi \cdot \sigma^2(f) \le \text{oce}(f) - R(f) \le \frac{\text{Lip}(\phi)}{2} \cdot \sigma(f) \tag{20}$$

$$C_\phi \cdot \sigma^2(f) \le R(f) - \overline{\text{oce}}(f) \le \frac{\text{Lip}(\phi)}{2} \cdot \sigma(f), \tag{21}$$

*where $C_\phi := \inf_{0 < |t| \le M} \frac{\phi(t) - t}{t^2} \ge 0$.*

We note that Gotoh et al. [20] also relates (dual forms of) OCE risks to variance, where the authors assume the twice continuous differentiability of the convex conjugate of the disutility function $\phi$ and use Taylor expansion to arrive at an asymptotic expression for the case $\text{Lip}(\phi) \to 1$. Lemma 6, on the other hand, exploits the convexity of $\phi$ and the dominance relations between disutility functions to provide nonasymptotic bound without requiring further smoothness assumptions on $\phi$. For example, the conjugate disutility function of CVaR is not differentiable, but Lemma 6 holds with $C_\phi = \frac{1}{M} \min\{1, \frac{\bar{\alpha}}{\alpha}\}$ and $\text{Lip}(\phi) = \frac{1}{\alpha}$.

Using Lemma 6, we can prove the following theorem (see Appendix A.7 for the proof).

**Theorem 7** (Expected loss guarantee). *Let $P$ be a fixed, unknown data-generating distribution, and let hypothesis space be bounded, i.e. there exists some $M > 0$ such that $\sup_{z \in \mathcal{Z}} f(z) \le M$ holds almost surely for all $f \in \mathcal{F}$. Then, for any $\delta \in (0, 1]$ and $n \ge 2$, we have*

$$R(\widehat{f}_{\text{eom}}) \le \left( R(f_{\text{avg}}^*) + \frac{\text{Lip}(\phi)}{2}\sigma(f_{\text{avg}}^*) \right) + 4\mathbf{E}[\mathfrak{R}(\mathcal{F}(Z^n))] + \frac{4M\sqrt{\log(3/\delta)}}{\sqrt{n}}, \tag{22}$$

*with probability at least $1 - \delta$. Under the same assumptions, we have*

$$R(\widehat{f}_{\text{eim}}) \le R(f_{\text{avg}}^*) + 4\mathbf{E}[\mathfrak{R}(\mathcal{F}(Z^n))] + \frac{4M\sqrt{\log(2/\delta)}}{\sqrt{n}} + \frac{\text{Lip}(\phi)}{2}\sigma_n(\widehat{f}_{\text{eim}}), \tag{23}$$

*with probability at least $1 - \delta$. Moreover, one can replace $f_{\text{avg}}^*$ in (22), (23) by any fixed $f \in \mathcal{F}$.*

In contrast to (18), the bound (22) is related to the disutility function only through a term proportional to $\sigma(f_{\text{avg}}^*)$. To see the benefit of this suppressed dependence, consider a case where the hypothesis space is a universal approximator (also known as the *realizable case*). Then, the first and second moment of loss population becomes zero, and Theorem 7 gives an ERM-like expected loss guarantee on the empirical OCE minimizer.

Regarding the bound for the EIM hypothesis, we remark that the non-vanishing term in the bound (23) depends on the behavior of the learned hypothesis on the training data only, unlike in (22); such discrepancy can help to recover ERM-like bounds under a milder assumption than universal approximability, for inverted OCE measures that make $\sigma_n(\widehat{f}_{\text{eim}})$ small (e.g., inverted entropic risk).

We remark that Lemma 6 indicates a potential connection of EOM to the sample variance penalization (SVP) procedure suggested by Maurer and Pontil [37]. Under suitable setups, one can show that the excess expected loss of the SVP hypothesis is $\mathcal{O}(1/n)$, even when the excess expected loss of ordinary ERM decays no faster than $1/\sqrt{n}$. An interesting open question is whether, and under what conditions, the EOM can provide a similar acceleration. Indeed, we observe that EOM provides a nontrivial acceleration under at least one specific scenario: the stylized example of [37].

**Example.** *Consider a hypothesis space consisting of only two hypotheses $\mathcal{F} = \{f_1, f_2\}$, such that under the presumed data-generating distribution $P$ we have*

$$f_1(Z) = \frac{1}{2}, \qquad f_2(Z) = \begin{cases} 0 & \cdots \text{ w.p. } \frac{1-\epsilon}{2} \\ 1 & \cdots \text{ w.p. } \frac{1+\epsilon}{2} \end{cases}, \tag{24}$$

*for some $\epsilon \in (0, \frac{1}{2})$. We are interested in the probability that EOM erroneously learns $f_2$ and incur the excess risk of size $\epsilon$. More formally, we aim to provide lower and upper bound on the excess risk probability as $\delta_{\text{eom}} := \mathbf{P}[\text{oce}_n(f_2) \leq \text{oce}_n(f_1)]$. If we focus on the case of CVaR, the excess risk probability becomes $\delta_{\text{eom}} = \mathbf{P}[X \leq \frac{n\alpha}{2}]$ where $X \sim \text{Bin}(n, \frac{1+\epsilon}{2})$, and analyze the binomial tail to give the following proposition (see Appendix A.8 for the proof).*

**Proposition 8** (Faster convergence). *There exists an absolute constant $C_1 > \frac{\sqrt{2}}{3}$, such that*

$$C_1 \exp\left(-4n\left(\epsilon + \bar{\alpha}\right)^2 - \log\sqrt{n\alpha} - \frac{16}{n}\right) \leq \delta_{\text{eom}} \leq \exp\left(-\frac{n(\epsilon + \bar{\alpha})^2}{2}\right) \tag{25}$$

*holds for the empirical CVaR minimizer with $\alpha \in (0, 1]$.*

*We observe that $\delta_{\text{eom}}$ can be made less than $\exp(-\frac{n}{2})$ by taking $\alpha \to 0$, regardless of $\epsilon$.*

# 4  Numerical simulations: Batch-SVP for CVaR minimization

Recall that Lemma 6 implies that the EOM can be relaxed to the SVP, where one aims to find

$$\widehat{f}_{\text{svp}} = \arg\min_{f \in \mathcal{F}} \left\{ R_n(f) + \lambda \cdot \sigma_n(f) \right\}, \tag{26}$$

for some hyperparameter $\lambda \geq 0$. At the same time, the relaxed form enjoys a favorable theoretical properties in terms of generalization [37], as briefly discussed in the previous section (although requiring a careful choice of $\lambda$). In light of this observation, we explore the potential benefit of using SVP as an additional *simple baseline* for algorithmic studies on OCE minimization, along with a popularly used baseline of batch-based EOM; batch-based empirical CVaR minimization (dubbed *batch-CVaR*) has been used as a baseline in recent algorithmic works on CVaR minimization [14, 48]. As will be shown shortly, we find that batch-based SVP (dubbed *batch-SVP*) can outperform batch-CVaR without an overly sophisticated selection of the hyperparameter $\lambda$.

**Setup.** We focus on the case of CVaR minimization on CIFAR-10 image classification task [29] where we use the standard cross-entropy loss. As a model, we use ResNet18 [24]. As an optimizer, we use Adam with weight decay [34] with a batch size 100 and PyTorch default learning rate. For CVaR, we have experimented with $\alpha = \{0.2, 0.4, 0.6, 0.8\}$. For batch-SVP, we have simply tested over $\lambda = \{0.5, 1.0\}$. All results are averaged over ten independent trials (more details at Appendix B).

**Results and discussion.** Trajectories of test and train CVaR for 100 epochs are given in Fig. 1 for $\alpha = \{0.2, 0.8\}$; plots for $\alpha = \{0.4, 0.6\}$ are given in Appendix B. We observe that batch-SVP

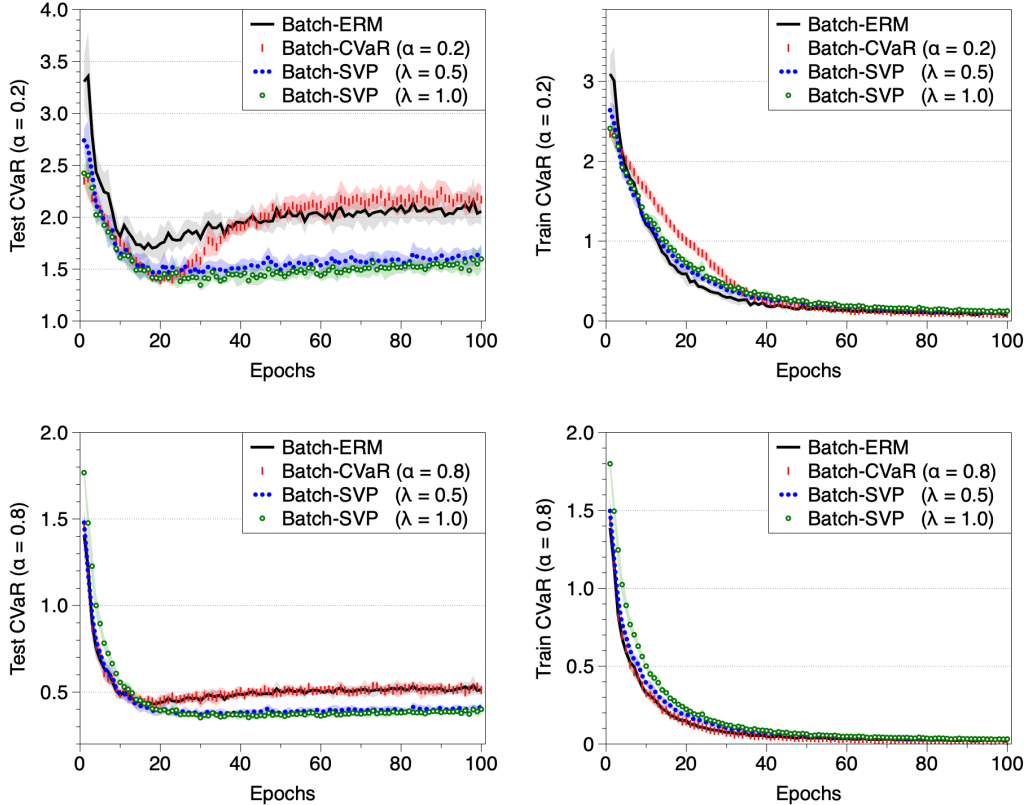

Figure 1: Trajectories of test/train CVaR (left/right) for hypotheses trained on ResNet18 and CIFAR-10 (Upper row: $\alpha = 0.2$, lower row: $\alpha = 0.8$). Shaded regions denote the (mean $\pm$ standard deviation) over ten independent trials.

hypotheses achieve a similar or better performance than batch-CVaR at the best epoch, and have a much more stable learning curve due to the regularization properties of SVP. Moreover, after $\sim 40$ epochs, batch-CVaR start to perform worse than vanilla ERM. Similar phenomenon has been reported by [14], where batch-CVaR (and even other sophisticated methods) underperform the vanilla ERM under a number of settings. The trajectories suggest that such CVaR optimization methods are suffering from over-training. SVP provides a baseline method, which does not have such issues.

**Code.** Available at `https://github.com/jaeho-lee/oce`.

## 5  Summary and future directions

In this paper, we have (a) presented general theoretical guarantees for risk-sensitive learning (Theorems 4 and 7), (b) established a new framework to study risk-seeking learning scheme (Section 2.2), and (c) rejuvenated the sample variance penalization as a baseline algorithm for risk-averse learning (Section 4). As future work, we aim to address the generalization properties of learning algorithms that simultaneously train a hypothesis and a *weighting function*, according to which the hypothesis will be evaluated [40, 46]. Formalizing such scenarios may accompany an investigation of the complexity (e.g., Rademacher averages, VC-dimension) of the space of all possible weighting functions based on a generalized notion of spectral risk measures [2].

## Broader impacts

This paper is focused on the subject of *risk-sensitivity*, which is a topic that is deeply intertwined with the safety, reliability, and fairness of machine intelligence (see, e.g., [42]). While our primary aim is to enhance theoretical understandings on the risk-sensitive learning, instead of proposing a new algorithm, we expect our results to have two broader consequences.

**Facilitating algorithmic advances.** For researchers trying to develop new risk-sensitive learning schemes, our general framework lowers the barrier to do so; we provide performance guarantees that applies for a broad class of algorithms that considers risk-seeking and risk-averse measures of loss (Theorems 4 and 7). Also, we provide a non-vacuous baseline to be compared with newly developed algorithms (Section 4). We believe that our theoretical framework will help stimulate further developments on risk-sensitive learning.

**Pondering on the cost of fairness.** One of our theoretical results (Theorem 4) can be interpreted as a characterization of (an instance of) the *cost of fairness* [13, 15, 38]. Indeed, recalling that CVaR is a fairness risk measure with an individual fairness criterion [53], Theorem 4 implies that the performance gap may grow wider if we try to apply a stricter fairness criterion. Such quantification of the cost of making a fairer decision is a double-edged sword; the cost may *scare* the decision-maker away from taking the fairness into account at all, or the cost may *guide* the decision-maker to find a fairest solution under the operational constraints. We sincerely hope that the latter is the case. Indeed, we also provide a result (Theorem 7) that the drawback of making a fair decision may not be big for modern machine learning applications!

## Acknowledgements

JL thanks Aolin Xu, Maxim Raginsky, Insu Han, Sungsoo Ahn, and Sihyun Yu for their helpful feedbacks on the early version of the manuscript. JL also acknowledges the comments from an anonymous NeurIPS reviewer, which helped us refine the constant for Lemma 3.

## Funding disclosure

This work was partly supported by Institute of Information & Communications Technology Planning & Evaluation (IITP) grant funded by the Korea government (MSIT, No.2019-0-00075, Artificial Intelligence Graduate School Program (KAIST)), and in part by the Engineering Research Center Program through the National Research Foundation of Korea (NRF) funded by the Korea Government (MSIT, NRF-2018R1A5A1059921).

## Footnotes

[2]We note that the most recent version of [14] (also appearing NeurIPS 2020) now provides an extension to the case of finite VC-dimension. Lemma 3 refines the extended result as well.

[3]We avoid using more popular terminologies ("risk" and "empirical risk") to prevent unnecessary confusion.

[4]We omit $P$ or $\phi$ in $\mathsf{oce}^\phi(f; P)$ when clear from context.

[5]Again, we drop $P$ whenever the choice is clear from context, and write $\sigma_n(f)$ for the empirical version.

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
