[Supplementary Material · supp.pdf]

# A Omitted proofs

## A.1 Proof of Proposition 1

We begin by plugging the disutility function $\phi(t) = \frac{n}{k}[t]_+$ to the definition of the inverted OCE risk (Definition 2) to get

$$\overline{\text{oce}}^\phi(f; P_n) = \sup_\lambda \left\{ \lambda - \frac{1}{k} \sum_{i=1}^n [\lambda - f(Z_i)]_+ \right\}. \tag{27}$$

While the optimum-achieving $\lambda$ may not be unique, we observe that $\lambda^* = f(Z_{\pi(k)})$ achieves the optimum. To see this, first observe that increasing $\lambda$ (non-strictly) increases the first term in the curly bracket ($\lambda$) and decreases the second term ($-\frac{1}{k} \sum_{i=1}^k [\lambda - f(Z_i)]_+$). By increasing $\lambda^*$ by $\Delta\lambda$, the second term decreases at least by $\Delta\lambda$, since at least $k$ terms among $\{\lambda - f(Z_i)\}_{i=1}^n$ are nonnegative. Likewise, by decreasing $\lambda^*$ by $\Delta\lambda$, the increment in the second term is no bigger than $\Delta\lambda$.

Plugging in $\lambda^* = f(Z_{\pi(k)})$, we get what we want.

## A.2 Proof of Proposition 2

We look at each inverted OCE separately.

*1. Inverted entropic risk:* From the elementary calculus, we know that

$$\overline{\text{oce}}(f) = -\frac{1}{\gamma} \log \mathbf{E}[e^{-\gamma f(Z)}], \qquad \lambda^* = -\frac{1}{\gamma} \log \mathbf{E}[e^{-\gamma f(Z)}]. \tag{28}$$

The corresponding influence function is then given as

$$\mathsf{IF}(z^*) = \frac{1}{\gamma} \cdot \lim_{\varepsilon \to 0^+} \frac{1}{\varepsilon} \log \left( \frac{\mathbf{E}[e^{-\gamma f(Z)}]}{\bar{\varepsilon} \cdot \mathbf{E}[e^{-\gamma f(Z)}] + \varepsilon \cdot e^{-\gamma f(z^*)}} \right) \tag{29}$$

$$= -\frac{1}{\gamma} \cdot \lim_{\varepsilon \to 0^+} \frac{1}{\varepsilon} \log \left( \bar{\varepsilon} + \varepsilon \frac{e^{-\gamma f(z^*)}}{\mathbf{E}[e^{-\gamma f(Z)}]} \right) \tag{30}$$

$$= \frac{1}{\gamma} - \frac{1}{\gamma} \frac{e^{-\gamma f(z^*)}}{\mathbf{E}[e^{-\gamma f(Z)}]}. \tag{31}$$

*2. Inverted mean-variance:* From the elementary calculus, we know that

$$\overline{\text{oce}}(f) = R(f) - c \cdot \sigma^2(f), \qquad \lambda^* = R(f). \tag{32}$$

The corresponding influence function is then given as

$$\mathsf{IF}(z^*) = \lim_{\varepsilon \to 0^+} \frac{1}{\varepsilon} \left[ -\varepsilon \cdot R(f) + \varepsilon \cdot f(z^*) + c\varepsilon \cdot \sigma^2(f) - c\varepsilon\bar{\varepsilon} \cdot (f(z^*) - R(f))^2 \right] \tag{33}$$

$$= f(z^*) - R(f) + c \left[ \sigma^2(f) - (f(z^*) - R(f))^2 \right]. \tag{34}$$

*3. Inverted CVaR:* Using the argument similar to [41], we get that

$$\overline{\text{oce}}(f) = \mathsf{q}(\alpha; f_{\sharp P}) - \frac{1}{\alpha} \mathbf{E}_P[\mathsf{q}(\alpha; f_{\sharp P}) - f(Z)]_+, \qquad \lambda^* = \mathsf{q}(\alpha; f_{\sharp P}). \tag{35}$$

We denote the $z^*$-contaminated distribution by $\tilde{P}$. Then, the corresponding influence function can be written as

$$\mathsf{IF}(z^*) = \lim_{\varepsilon \to 0^+} \frac{1}{\varepsilon} \Big[ \mathsf{q}(\alpha; f_{\sharp\tilde{P}}) - \mathsf{q}(\alpha; f_{\sharp P}) + \frac{1}{\alpha} \mathbf{E}_P[\mathsf{q}(\alpha; f_{\sharp P}) - f(Z)]_+$$

$$- \frac{\bar{\varepsilon}}{\alpha} \mathbf{E}_P[\mathsf{q}(\alpha; f_{\sharp\tilde{P}}) - f(Z)]_+ - \frac{\varepsilon}{\alpha} [\mathsf{q}(\alpha; f_{\sharp\tilde{P}}) - f(z^*)]_+ \Big]. \tag{36}$$

By adding and subtracting $\frac{\bar{\varepsilon}}{\alpha}\mathbf{E}_P[q(\alpha; f_{\sharp P}) - f(Z)]_+$, we can rewrite as

$$= \frac{1}{\alpha}\mathbf{E}_P[\mathfrak{q}(\alpha; f_{\sharp P}) - f(Z)]_+ - \frac{1}{\alpha}[\mathfrak{q}(\alpha; f_{\sharp P}) - f(z^*)]_+$$
$$+ \lim_{\varepsilon \to 0^+} \frac{1}{\varepsilon}\left[\mathfrak{q}(\alpha; f_{\sharp \tilde{P}}) - \mathfrak{q}(\alpha; f_{\sharp P}) + \frac{\bar{\varepsilon}}{\alpha}\mathbf{E}_P\left[[\mathfrak{q}(\alpha; f_{\sharp P}) - f(Z)]_+ - [\mathfrak{q}(\alpha; f_{\sharp \tilde{P}}) - f(Z)]_+\right]\right].$$
(37)

Now, notice that the limiting term is 0 whenever whenever $f_{\sharp P}$ does not have a point mass around $\mathfrak{q}(\alpha; f_{\sharp P})$. Indeed, $[\mathfrak{q}(\alpha; f_{\sharp P}) - f(Z)]_+ - [\mathfrak{q}(\alpha; f_{\sharp \tilde{P}}) - f(Z)]_+$ is zero with probability $1 - \alpha$, and $\mathfrak{q}(\alpha; f_{\sharp P}) - \mathfrak{q}(\alpha; f_{\sharp \tilde{P}})$ with probability $\alpha$. Taking the limit, remaining terms cancel out.

### A.3 Proof of Lemma 3

We begin by giving the following technical lemma.

**Lemma 9.** *Suppose that $P, f$ satisfies $f(Z) \in [0, M]$ almost surely for $Z \sim P$. Then, we have*

$$\mathsf{oce}(f) = \min_{\lambda \in [0,M]} \left\{\lambda + \mathbf{E}_P \phi(f(Z) - \lambda)\right\}$$
(38)

$$\overline{\mathsf{oce}}(f) = \max_{\lambda \in [0,M]} \left\{\lambda - \mathbf{E}_P \phi(\lambda - f(Z))\right\}.$$
(39)

*Proof.* See Appendix A.9. $\square$

In other word, the search space of the variational parameter $\lambda$ appearing in the definition of OCE risk (3) can be constrained to a finite length interval, given that the random variable $f(Z)$ is also bounded. Using this result, we can take a closer look at the one-sided deviation; for any $f \in \mathcal{F}$, we have

$$\mathsf{oce}(f) - \mathsf{oce}_n(f) = \min_{\lambda \in [0,M]} \left\{\lambda + \mathbf{E}_P \phi(f(Z) - \lambda)\right\} - \min_{\lambda \in [0,M]} \left\{\lambda + \mathbf{E}_{P_n} \phi(f(Z) - \lambda)\right\} \quad (40)$$

$$\leq \max_{\lambda \in [0,M]} \left\{\mathbf{E}_P \phi(f(Z) - \lambda) - \mathbf{E}_{P_n} \phi(f(Z) - \lambda)\right\}, \quad (41)$$

where the inequality holds by selecting the first $\lambda$ to be identical to the second $\lambda$. Taking supremum over $\mathcal{F}$ on both sides, we get

$$\sup_{f \in \mathcal{F}} \left\{\mathsf{oce}(f) - \mathsf{oce}_n(f)\right\} \leq \sup_{g \in \mathcal{G}} \left\{\mathbf{E}_P[\phi \circ g(Z)] - \mathbf{E}_{P_n}[\phi \circ g(Z)]\right\}, \quad (42)$$

where $\mathcal{G} := \{f(\cdot) - \lambda \mid f \in \mathcal{F}, \lambda \in [0, M]\}$ is a product hypothesis space constructed upon $\mathcal{F}$ and $[0, M]$. To bound the (one-sided) uniform deviation, we first control its expectation via Rademacher averages. As $\phi(0) = 0$ holds by definition, the contraction principle (see, e.g., [31, Eq. 4.20]) gives

$$\mathbf{E} \sup_{g \in \mathcal{G}} \left\{\mathbf{E}_P[\phi \circ g(Z)] - \mathbf{E}_{P_n}[\phi \circ g(Z)]\right\} \leq 2\mathrm{Lip}(\phi) \cdot \mathbf{E}\mathfrak{R}_n(\mathcal{G}(Z^n)). \quad (43)$$

Now, the Rademacher average of $\mathcal{G}$ can be bounded as

$$\mathbf{E}\mathfrak{R}_n(\mathcal{G}) = \mathbf{E}\mathbf{E}_{\epsilon^n} \sup_{\substack{f \in \mathcal{F}, \\ \lambda \in [0,M]}} \left(\frac{1}{n}\sum_{i=1}^n \epsilon_i f(Z_i) - \frac{1}{n}\sum_{i=1}^n \epsilon_i \lambda\right) \quad (44)$$

$$\leq \mathbf{E}\mathbf{E}_{\epsilon^n} \sup_{f \in \mathcal{F}} \left(\frac{1}{n}\sum_{i=1}^n \epsilon_i f(Z_i)\right) + \mathbf{E}_{\epsilon^n} \sup_{\lambda \in [0,M]} \left(\lambda \cdot \frac{1}{n}\sum_{i=1}^n \epsilon_i\right) \quad (45)$$

$$\leq \mathbf{E}\mathfrak{R}(\mathcal{F}(Z^n)) + \frac{M}{n} \cdot \mathbf{E}_{\epsilon^n} \left|\sum_{i=1}^n \epsilon_i\right| \quad (46)$$

$$\leq \mathbf{E}\mathfrak{R}(\mathcal{F}(Z^n)) + \frac{M}{\sqrt{n}}, \quad (47)$$

where the last line follows from the Jensen's inequality $(\mathbf{E}|X|)^2 \leq \mathbf{E}[|X|^2] = \mathbf{E}[X^2]$. Combining (47) with (43), we get

$$\mathbf{E} \sup_{g \in \mathcal{G}} \left\{ \mathbf{E}_P[\phi \circ g(Z)] - \mathbf{E}_{P_n}[\phi \circ g(Z)] \right\} \leq \mathrm{Lip}(\phi) \cdot \left( 2\mathbf{E}\mathfrak{R}(\mathcal{F}(Z^n)) + \frac{2M}{\sqrt{n}} \right) \quad (48)$$

Combining with the McDiarmid's inequality to control the residual term, we have

$$\sup_{g \in \mathcal{G}} \left\{ \mathbf{E}_P[\phi \circ g(Z)] - \mathbf{E}_{P_n}[\phi \circ g(Z)] \right\} \quad (49)$$

$$\leq \mathrm{Lip}(\phi) \left( 2\mathbf{E}\mathfrak{R}_n(\mathcal{F}(Z^n)) + \frac{M(2 + \sqrt{\log(2/\delta)})}{\sqrt{n}} \right) \quad \text{w.p. } 1 - \frac{2}{\delta}. \quad (50)$$

The other direction can be derived similarly. Using the union bound, we get what we want.

To get the same bound with $\overline{\mathsf{oce}}$, we slightly modify Eqs. (40) and (41) as follows.

$$\overline{\mathsf{oce}}(f) - \overline{\mathsf{oce}}_n(f) = \max_{\lambda \in [0,M]} \left\{ \lambda - \mathbf{E}_P \phi(\lambda - f(Z)) \right\} - \max_{\lambda \in [0,M]} \left\{ \lambda - \mathbf{E}_{P_n} \phi(\lambda - f(Z)) \right\} \quad (51)$$

$$\leq \max_{\lambda \in [0,M]} \left\{ \mathbf{E}_{P_n} \phi(\lambda - f(Z)) - \mathbf{E}_P \phi(\lambda - f(Z)) \right\}, \quad (52)$$

where the inequality holds by selecting the second $\lambda$ to be equal to the first $\lambda$. The remaining steps are identical to the proof of the claim for oce.

## A.4 Proof of Theorem 4

The claim is a direct consequence of Lemma 3. Indeed, we can proceed as

$$\mathsf{oce}(\widehat{f}_{\mathsf{eom}}) - \mathsf{oce}(f_{\mathsf{oce}}^*)$$
$$= \left[ \mathsf{oce}(\widehat{f}_{\mathsf{eom}}) - \mathsf{oce}_n(\widehat{f}_{\mathsf{eom}}) \right] + \underbrace{\left[ \mathsf{oce}_n(\widehat{f}_{\mathsf{eom}}) - \mathsf{oce}_n(f_{\mathsf{oce}}^*) \right]}_{\leq 0} + \left[ \mathsf{oce}_n(f_{\mathsf{oce}}^*) - \mathsf{oce}(f_{\mathsf{oce}}^*) \right], \quad (53)$$

where the nonpositivity of second term follows from the definition of $\widehat{f}_{\mathsf{eom}}$. The remaining terms can be bounded via Lemma 3 to get the claimed result.

The proof for $\widehat{f}_{\mathsf{eim}}$ can be done equivalently, by

$$\overline{\mathsf{oce}}(\widehat{f}_{\mathsf{eim}}) - \overline{\mathsf{oce}}(f_{\overline{\mathsf{oce}}}^*)$$
$$= \left[ \overline{\mathsf{oce}}(\widehat{f}_{\mathsf{eom}}) - \overline{\mathsf{oce}}_n(\widehat{f}_{\mathsf{eom}}) \right] + \underbrace{\left[ \overline{\mathsf{oce}}_n(\widehat{f}_{\mathsf{eom}}) - \overline{\mathsf{oce}}_n(f_{\overline{\mathsf{oce}}}^*) \right]}_{\leq 0} + \left[ \overline{\mathsf{oce}}_n(f_{\overline{\mathsf{oce}}}^*) - \overline{\mathsf{oce}}(f_{\overline{\mathsf{oce}}}^*) \right], \quad (54)$$

## A.5 Proof of Proposition 5

To get $0 \leq \overline{\mathsf{oce}}(f)$, plug in $\lambda = 0$ to Definition 2 to observe that

$$\overline{\mathsf{oce}}(f) \geq -\mathbf{E}_P \phi(-f(Z)) \geq -\mathbf{E}_P \phi(0) = 0, \quad (55)$$

where the second inequality holds as $\phi$ is a nondecreasing function.

To get $\overline{\mathsf{oce}}(f) \leq R(f)$, we first observe that $\phi(t) \geq t$ holds for all $t$, as $\phi$ is a convex function with $\phi(0) = 0, 1 \in \partial\phi(0)$. Thus, we get

$$\overline{\mathsf{oce}}(f) \leq \sup_{\lambda \in \mathbb{R}} \left\{ \lambda - \mathbf{E}_P[\lambda - f(Z)] \right\} = \mathbf{E}_P f(Z) = R(f). \quad (56)$$

To get $R(f) \leq \mathsf{oce}(f)$, use again that $\phi(t) \geq t$ to proceed as

$$\inf_{\lambda \in \mathbb{R}} \left\{ \lambda + \mathbf{E}_P \phi(f(Z) - \lambda) \right\} \geq \inf_{\lambda \in \mathbb{R}} \left\{ \lambda + \mathbf{E}_P[f(Z) - \lambda] \right\} = R(f). \quad (57)$$

To get $\mathsf{oce}(f) \leq \mathrm{Lip}(\phi) \cdot R(f)$, observe that

$$\inf_{\lambda \in \mathbb{R}} \left\{ \lambda + \mathbf{E}_P \phi(f(Z) - \lambda) \right\} \leq \mathbf{E}_P \phi(f(Z)) \leq \mathrm{Lip}(\phi) \cdot \mathbf{E}_P |f(Z) - 0| = \mathrm{Lip}(\phi) \cdot R(f), \quad (58)$$

where for the first inequality we plugged in the special case $\lambda = 0$.

## A.6 Proof of Lemma 6

We first prove the bounds for oce. To get the lower bound, we start from Lemma 9 and proceed as

$$\inf_{\lambda \in [0,M]} \left\{ \lambda + \mathbf{E}_P \phi(f(Z) - \lambda) \right\} \geq \inf_{\lambda \in [0,M]} \left\{ \lambda + \mathbf{E}_P[f(Z) - \lambda] + C_\phi \cdot \mathbf{E}_P(f(Z) - \lambda)^2 \right\} \quad (59)$$

$$= R(f) + C_\phi \cdot \inf_{\lambda \in [0,M]} \mathbf{E}_P(f(Z) - \lambda)^2 \quad (60)$$

$$= R(f) + C_\phi \cdot \sigma^2(f), \quad (61)$$

where the inequality holds by the definition of $C_\phi$. To get the upper bound, plug in $\lambda = R(f)$ to the variational definition (3) and observe that

$$\mathsf{oce}(f) \leq R(f) + \mathbf{E}_P \phi(f(Z) - R(f)) \quad (62)$$

$$\leq R(f) + \mathrm{Lip}(\phi) \cdot \mathbf{E}_P[f(Z) - R(f)]_+ \quad (63)$$

$$= R(f) + \frac{\mathrm{Lip}(\phi)}{2} \cdot \mathbf{E}_P|f(Z) - R(f)| \quad (64)$$

$$\leq R(f) + \frac{\mathrm{Lip}(\phi)}{2} \cdot \sqrt{\mathbf{E}_P(f(Z) - R(f))^2}, \quad (65)$$

where the last line holds by the Jensen's inequality.

The bounds for $\overline{\mathsf{oce}}$ can be proved similarly. For the lower bound, we plug in $\lambda = R(f)$ to get

$$\overline{\mathsf{oce}}(f) \geq R(f) - \mathbf{E}_P \phi(R(f) - f(Z)) \quad (66)$$

$$\geq R(f) - \mathrm{Lip}(\phi) \cdot \mathbf{E}_P|R(f) - f(Z)|_+ \quad (67)$$

$$\geq R(f) - \frac{\mathrm{Lip}(\phi)}{2} \sigma(f). \quad (68)$$

For the upper bound, we use the definition of $C_\phi$ to get

$$\sup_{\lambda \in [0,M]} \left\{ \lambda - \mathbf{E}_P \phi(\lambda - f(Z)) \right\} \leq \sup_{\lambda \in [0,M]} \left\{ \lambda - \mathbf{E}_P[\lambda - f(Z)] - C_\phi \cdot \mathbf{E}_P(\lambda - f(Z))^2 \right\} \quad (69)$$

$$= R(f) - C_\phi \cdot \inf_{\lambda \in [0,M]} \mathbf{E}_P(\lambda - f(Z))^2 \quad (70)$$

$$= R(f) - C_\phi \cdot \sigma^2(f) \quad (71)$$

## A.7 Proof of Theorem 7

To prove the first claim, we start with Lemma 6 to proceed as

$$R(\widehat{f}_{\mathsf{eom}}) \leq R_n(\widehat{f}_{\mathsf{eom}}) + \sup_{f \in \mathcal{F}} |R(f) - R_n(f)| \quad (72)$$

$$\leq \mathsf{oce}_n(\widehat{f}_{\mathsf{eom}}) - C_\phi \sigma_n^2(\widehat{f}_{\mathsf{eom}}) + \sup_{f \in \mathcal{F}} |R(f) - R_n(f)| \quad (73)$$

$$\leq \mathsf{oce}_n(f_{\mathsf{avg}}^*) - C_\phi \sigma_n^2(\widehat{f}_{\mathsf{eom}}) + \sup_{f \in \mathcal{F}} |R(f) - R_n(f)| \quad (74)$$

$$\leq R_n(f_{\mathsf{avg}}^*) + \frac{\mathrm{Lip}(\phi)}{2} \sigma_n(f_{\mathsf{avg}}^*) - C_\phi \sigma_n^2(\widehat{f}_{\mathsf{eom}}) + \sup_{f \in \mathcal{F}} |R(f) - R_n(f)| \quad (75)$$

$$\leq R(f_{\mathsf{avg}}^*) + \frac{\mathrm{Lip}(\phi)}{2} \sigma_n(f_{\mathsf{avg}}^*) - C_\phi \sigma_n^2(\widehat{f}_{\mathsf{eom}}) + 2 \sup_{f \in \mathcal{F}} |R(f) - R_n(f)|. \quad (76)$$

As $f_{\mathsf{avg}}^*$ is a fixed object independent of the samples (for given $P$), Theorem 10 of [37] implies

$$\sigma_n(f_{\mathsf{avg}}^*) - \sigma(f_{\mathsf{avg}}^*) \leq M \sqrt{\frac{2 \log(3/\delta)}{n-1}} \leq 2M \sqrt{\frac{\log(3/\delta)}{n}}, \qquad \text{w.p. } 1 - \frac{\delta}{3}. \quad (77)$$

Combining with standard symmetrization bounds on uniform deviation (see, e.g., [45]) with excess risk probability $2\delta/3$, we get the first claim.

The second claim can be proved similarly, but without invoking the concentration of sample variance. We proceed as follows.

$$R(\widehat{f}_{\mathsf{eim}}) \le R_n(\widehat{f}_{\mathsf{eim}}) + \sup_{f \in \mathcal{F}} |R(f) - R_n(f)| \tag{78}$$

$$\le \overline{\mathsf{oce}}_n(\widehat{f}_{\mathsf{eom}}) + \mathrm{Lip}(\phi) \cdot \sigma_n(\widehat{f}_{\mathsf{eom}}) + \sup_{f \in \mathcal{F}} |R(f) - R_n(f)| \tag{79}$$

$$\le \overline{\mathsf{oce}}_n(f_{\mathsf{avg}}^*) + \mathrm{Lip}(\phi) \cdot \sigma_n(\widehat{f}_{\mathsf{eom}}) + \sup_{f \in \mathcal{F}} |R(f) - R_n(f)| \tag{80}$$

$$\le R_n(f_{\mathsf{avg}}^*) + \mathrm{Lip}(\phi) \cdot \sigma_n(\widehat{f}_{\mathsf{eom}}) + \sup_{f \in \mathcal{F}} |R(f) - R_n(f)| \tag{81}$$

$$\le R(f_{\mathsf{avg}}^*) + \mathrm{Lip}(\phi) \cdot \sigma_n(\widehat{f}_{\mathsf{eom}}) + 2 \sup_{f \in \mathcal{F}} |R(f) - R_n(f)|. \tag{82}$$

Plugging in the standard Rademacher average bound, we get what we want.

## A.8  Proof of Proposition 8

We begin by observing that $\delta_{\mathsf{eom}}$ for CVaR reduces to the binomial tail probability.

**Lemma 10.** *Let $\delta_{\mathsf{eom}}$ be the excess risk probability of empirical CVaR minimization for some $\alpha \in (0, 1)$. Then, we have $\delta_{\mathsf{eom}} = \mathbf{P}\big[X \le \frac{n\alpha}{2}\big]$, where $X \sim \mathrm{Bin}\big(n, \frac{1+\epsilon}{2}\big)$.*

*Proof.* See Appendix A.10. □

To get the upper bound, observe that Lemma 10 can also be stated in the following form: the excess risk probability of the empirical CVaR minimization is

$$\delta_{\mathsf{eom}} = \mathbf{P}\left[\frac{1}{n} \sum_{i=1}^{n} X_i \le \frac{\alpha}{2}\right], \qquad \text{where } X_i \sim \mathrm{Bern}\left(\frac{1+\epsilon}{2}\right). \tag{83}$$

Then, the upper bound follows from the Hoeffding's inequality.

For the lower bound, we bound the binomial tail from below by the largest term in the binomial sum. Using the Stirling's approximation, we have for any $k \le n$,

$$\mathbf{P}\left[X \le k\right] \ge \binom{n}{k} \left(\frac{1+\epsilon}{2}\right)^k \left(\frac{1-\epsilon}{2}\right)^{n-k} \ge \sqrt{\frac{2\pi n}{e^4 k(n-k)}} \exp\left(-n\mathfrak{d}\left(\frac{k}{n}\Big\|\frac{1+\epsilon}{2}\right)\right), \tag{84}$$

where $\mathfrak{d}(p\|q)$ denotes the binary Kullback-Leibler (KL) divergence. To complete the lower bound, we note the quadratic upper and lower bound on the binary KL divergence.

**Lemma 11.** *For any $p, q \in (0, 1)$, we have $\mathfrak{d}(p\|q) \ge 2(p - q)^2$. If we further assume $q \in \big(\frac{1}{2}, \frac{3}{4}\big)$, then we also have $\mathfrak{d}(p\|q) \le 8(p - q)^2$.*

*Proof.* See Appendix A.11. □

From Lemma 11, the binary KL divergence of our interest can be upper bounded as

$$\mathfrak{d}\left(\frac{\lfloor \frac{n\alpha}{2} \rfloor}{n}\Big\|\frac{1+\epsilon}{2}\right) \le 8\left(\frac{1+\epsilon}{2} - \frac{\lfloor \frac{n\alpha}{2} \rfloor}{n}\right)^2 \le 8\left(\frac{\epsilon + \bar{\alpha}}{2} + \frac{1}{n}\right)^2 \le 4(\epsilon + \bar{\alpha})^2 + \frac{16}{n^2}, \tag{85}$$

where the last inequality holds by the Jensen's inequality. Combining Eq. (85) with Eq. (84), we can proceed as

$$\mathbf{P}\left[X \le \frac{n\alpha}{2}\right] \ge \sqrt{\frac{2\pi n}{e^4 \lfloor \frac{n\alpha}{2} \rfloor (n - \lfloor \frac{n\alpha}{2} \rfloor)}} \exp\left(-4n(\epsilon + \bar{\alpha})^2 - \frac{16}{n}\right) \tag{86}$$

$$\ge \sqrt{\frac{4\pi}{e^4}} \exp\left(-4n(\epsilon + \bar{\alpha})^2 - \frac{16}{n} - \log\sqrt{n\alpha}\right). \tag{87}$$

## A.9 Proof of Lemma 9

We first prove Eq. (38). For simplicity, we use the shorthand notation

$$\zeta(\lambda) := \lambda + \mathbf{E}_P \phi(f(Z) - \lambda). \tag{88}$$

Note that $\zeta$ is a continuous function, as the convexity of $\phi$ implies the convexity (and continuity) of $\mathbf{E}\phi$. We prove the claim by contradiction; for any (supposedly optimal) $\lambda^* \notin [0, M]$, we show that there exists a corresponding $\tilde{\lambda} \in [0, M]$ such that $\zeta(\tilde{\lambda}) \leq \zeta(\lambda^*)$.

**Case ($\lambda^* > M$).** We show that $\zeta(M+\varepsilon) \geq \zeta(M)$ for any $\varepsilon > 0$. Indeed, by considering a negative random variable $X := f(Z) - M$ lying in the interval $[-M, 0]$, we get

$$\zeta(M + \varepsilon) = M + \varepsilon + \mathbf{E}\phi(X - \varepsilon) \geq M + \varepsilon + \mathbf{E}\phi(X) - \varepsilon = \zeta(M), \tag{89}$$

where the inequality holds because $\phi$ is a convex function having 1 as a subgradient at 0.

**Case ($\lambda^* < 0$).** Similarly, we show $\zeta(-\varepsilon) \geq \zeta(0)$ for any $\varepsilon > 0$. By considering a positive random variable $X := f(Z)$, we get

$$\zeta(-\varepsilon) = -\varepsilon + \mathbf{E}\phi(X + \varepsilon) \geq \mathbf{E}\phi(X) = \zeta(0), \tag{90}$$

where the inequality follows from the fact that $\phi$ is a convex function having 1 as a subgradient at 0.

With $\zeta$ being a continuous function and $[0, M]$ being a compact search space, we can replace the infimum by minimum. Eq. (39) can be proved in a similar manner.

## A.10 Proof of Lemma 10

We begin by introducing the shorthand notation $X := \sum_{i=1}^{n} f_2(Z_i)$. From the setup, we know that $X \sim \text{Bin}(n, \frac{1+\epsilon}{2})$. From Lemma 9, we can proceed as

$$\text{oce}_n(f_2) = \min_{\lambda \in [0,1]} \left\{ \lambda + \frac{1}{n\alpha} \sum_{i=1}^{n} [f(Z_i) - \lambda]_+ \right\} \tag{91}$$

$$= \min_{\lambda \in [0,1]} \left\{ \lambda + \frac{1 - \lambda}{n\alpha} X \right\} \tag{92}$$

$$= \min \left\{ 1, \frac{X}{n\alpha} \right\}. \tag{93}$$

Thus, $\text{oce}_n(f_2) \leq \text{oce}_n(f_1) = \frac{1}{2}$ holds if and only if $X \leq \frac{n\alpha}{2}$.

## A.11 Proof of Lemma 11

To get the lower bound, we view $\mathfrak{d}(p\|q)$ as a function of $p$ and use the Taylor's theorem. The partial derivatives of the binary KL divergence with respect to $p$ are as follows.

$$\frac{\partial \mathfrak{d}(p\|q)}{\partial p} = \log \frac{p\bar{q}}{\bar{p}q}, \qquad \frac{\partial^2 \mathfrak{d}(p\|q)}{\partial p^2} = \frac{1}{p} + \frac{1}{\bar{p}}. \tag{94}$$

Note that as $p \in (0, 1)$, the second derivative is bounded from below by 4. Evaluating $\mathfrak{d}(\cdot\|q)$ at $q$, we have for some $p^*$ in the interval between $p$ and $q$,

$$\mathfrak{d}(p\|q) = \mathfrak{d}(q\|q) + \frac{\partial \mathfrak{d}(\cdot\|q)}{\partial p}(q)(p - q) + \frac{1}{2} \frac{\partial \mathfrak{d}(\cdot\|q)}{\partial p}(p^*)(p - q)^2 \geq 2(p - q)^2. \tag{95}$$

To get the upper bound, we view $\mathfrak{d}(p\|q)$ as a function of $q$ and use the Taylor's theorem again. The partial derivatives with respect to $q$ are

$$\frac{\partial \mathfrak{d}(p\|q)}{\partial q} = \frac{\bar{p}}{\bar{q}} - \frac{p}{q}, \qquad \frac{\partial^2 \mathfrak{d}(p\|q)}{\partial q^2} = \frac{\bar{p}}{\bar{q}^2} + \frac{p}{q^2}. \tag{96}$$

Given that $q \in (\frac{1}{2}, \frac{3}{4})$, we know that the second derivative is uniformly upper bounded as

$$\frac{\bar{p}}{\bar{q}^2} + \frac{p}{q^2} \leq 16\bar{p} + 4p \leq 16. \tag{97}$$

Evaluating $\mathfrak{d}(p\|\cdot)$ at $p$ in the same manner as Eq. (95), we get the upper bound.

Figure 2: Trajectories of test/train CVaR (left/right) for hypotheses trained on ResNet18 and CIFAR-10 ($\alpha = \{0.4, 0.6\}$).

## B  Additional plots and other experimental details

We now provide extra experimental details that are not given in Section 4. Unless stated otherwise, we follow PyTorch default parameters. One may also find our (primitive) PyTorch-based implementation at the following URL: https://github.com/jaeho-lee/oce.

**Dataset.**  We use CIFAR-10 image classification dataset [29], normalized using the constants $(0.4914, 0.4822, 0.4465), (0.247, 0.243, 0.261)$. We used random cropping (with padding of 4) and random horizontal flipping for augmentation.

**Optimization.**  We use mini-batch gradient descent, i.e. sampling without replacement until every samples are drawn.

# C    Related work

Here, we give a slightly extended overview of the related work, in addition to what has been already introduced in the main text. In particular, we focus on the following three topics: optimization of OCE risks, comparisons with another risk-sensitivity framework, and connections to the algorithmic fairness literature.

**Optimization of OCE.** The OCE risk measures belong to a wider class of convex risk measures [18], which was originally proposed as a relaxation of the notion of coherent risk measures [3]. Under classic setups equipped with convexity assumptions on the loss function, the fact that "the composite function of convex functions are also convex" enables one to use standard optimization techniques developed for the expected loss. In modern machine learning applications which accompanies batch-based nonconvex optimization, the optimization can be done with some additional tricks. In [14], the authors propose to use DPP-based techniques for a more accurate estimation of the conditional value-at-risk. In [33], the authors give a stochastic optimization algorithm which often outperforms the batch-based version.

**Comparison with rank-based measures.** The utility-theoretic framework of OCE risks (and their inverses) is complementary to another class of risk measures revolving around the quantile function of the loss population. Known as *spectral risk measures* [2] in the financial mathematics and as *L-statistics* (see, e.g., [26, 27]) in the statistics literature, the quantile-based approach focuses on the risk measures that can be written as

$$M_\psi(f, P) = \int_0^1 \psi(t) \cdot \mathfrak{q}(t; f_\sharp P) \mathrm{d}t, \tag{98}$$

for some *weighting function* $\phi : [0, 1] \to \mathbb{R}$ (satisfying varying degree of assumptions). While two frameworks share some commonalities (e.g., having the conditional value-at-risk as its special case), there is a subtle yet important difference: the utility-based framework allows the relative weight of the samples to depend on the *loss value* itself, while the quantile-based framework does not. In this sense, the OCE framework can be viewed as having a little more room for adaptation with respect to different distributions of loss. On the other hand, it is also true that the quantile-based framework covers some risk measures that are not describable via the OCE framework, e.g., risk measures trimming both samples with small and large loss values. An in-depth comparative study on the theoretical and empirical benefits of two frameworks may be an interesting direction of future study.

**Connections to algorithmic fairness.** In [53], Williamson and Menon give an axiomatic definition of *fairness risk measures* for group fairness. In particular, they argue that the fairness risk measure should be convex, positively homogeneous, monotonic, lower semi-continuous, translation invariant, averse, and law invariant. From the axioms, the authors propose a fairness-aware objective based on minimizing the conditional value-at-risk, which is a special case of the OCE risk with the disutility function $\phi(t) = \frac{1}{\alpha}[t]_+$. Indeed, the conditional value-at-risk can be simply viewed as a solution of

$$\max_{\substack{\mathcal{U} \subset \mathcal{Z} \\ P(\mathcal{U}) = \alpha}} \mathbf{E}_P[f(Z)| Z \in \mathcal{U}] \tag{99}$$

(given that $P$ has a density on $\mathcal{Z}$), which is the *worst-case* subgroup error over all subgroups of fraction $\alpha$. In another concurrent work [33], Li et al. also empirically observe that minimizing the entropic risk (instead of the expected loss) mitigates the disparate impact of the learned hypothesis on subgroups.