[Reviews · NeurIPS 2020]

Review 1

Summary and Contributions: This paper provides general theoretical guarantees for risk-sensitive learning. In particular, they study various risk sensitive measures under a generalized framework i.e. the optimized certainty equivalent risk. For the empirical OCE minimizer they provide bounds on the excess OCE risk and the excess expected loss. These bounds are in term of the Rademacher complexity of the hypothesis space. They also formalize risk-seeking learning.

Strengths: The study of risk sensitive (averse or seeking) is one of importance and relevance to the machine learning community. As such theoretical bounds on the excess expected loss and OCE risk for the OCE minimizer is a useful contribution. Moreover, these bounds are provided in terms of a data dependent measure of the hypothesis space i.e. Rademacher complexity.

Weaknesses: There are few limitations of the current paper: It is not clear how to compute the empirical OCE minimizer that the paper provides bounds for. In contrast, the papers cited as related work, especially Soma et. al. [2020], study the convergence properties of the optimum found by the SGD algorithm which seems more applicable to the machine learning community. In particular this paper does not address the question of optimization of the objective function even though it claims that the same has nice properties such as convexity. The paper also claims that there is prior theoretical work on stability and convergence of the minimization of risk-averse learning objective, this work has been incorporated in the current one. It leaves the picture of risk sensitive learning incomplete. -------------------------------------------------------- Edit ---------------------------------------------- My concerns were sufficiently addressed by the rebuttal and I tend toward keeping my score. ------------------------------------------------------------------------------------------------------------

Correctness: I have skimmed the proofs provided in the supplementary and they seem sound to me.

Clarity: The paper is well written and organized.

Relation to Prior Work: The paper cites only three references as ones directly related to the current work i.e. [12], [14], [44]. It is not clear whether this is an exhaustive list. Moreover, it is mentioned that these works differ from the current one for the framework they study. It would be nice to elaborate on this especially in terms of putting these works into the framework of the current paper for a better direct comparison of the results.

Reproducibility: Yes

Additional Feedback:


Review 2

Summary and Contributions: The paper analyzes generalization bounds the setting of what they refer to as risk sensitive loss functions. They introduce the notion of inverted optimized certainty equivalence to capture the idea that training of machine learning classifiers should sometimes focus on samples with the lowest possible loss. On a technical level, the authors leverage uniform convergence arguments based on the Rademacher complexity of the function class in question in order to get upper bounds on the generalization error of their empirical estimates. Furthermore, they experimentally evaluate their ideas on image recognition tasks.

Strengths: The claims in the paper are well substantiated and the overall problem of examining risk sensitive objectives seems interesting.

Weaknesses: I believe that the main weakness of the paper is that on a technical level, the results (lemma 2 and theorem 3) are just direct extensions of classic uniform convergence arguments based on rademacher complexity. Once you assume that the function class is uniformly bounded, then all the classic Rademacher complexity arguments go through. In that sense, there doesn’t seem to be anything particular about the fact that these are “risk sensitive” objectives. Also, at multiple points in the paper, the authors allude to the behavior of these generalization bound in the context of DNNs where one might hope for realizability assumptions to hold (see L208 , 245-247). Any kinds of arguments in this setting seem vacuous if one cannot control the rademacher complexity of the function class. Furthermore at several points in the paper, the authors attempt to connect their results to the fair ML literature, but these were never explicitly spelled out in detail in the main body of the paper.

Correctness: Yes, the theoretical results seem correct.

Clarity: The quality of the prose is clear.

Relation to Prior Work: There is no stand alone related work section, although connections to the literature are discussed in the introduction. The paper could however benefit by further discussing connections to the statistical learning theory literature and how their bounds differ from those typically found therein.

Reproducibility: Yes

Additional Feedback: ===== Updates ===== After further discussions with the other reviewers, I have decided to revise my score.


Review 3

Summary and Contributions: This paper studies the generalization properties of risk-averse and risk-seeking risk measures through optimized certainty equivalents (OCE). In particular, risk-seeking behavior is achieved through inverting the OCEs. The paper provides two types of results: bounds on the OCE via uniform convergence and bounds on the usual risk (expected loss) via bounds on the OCE and a variance argument. Some experiments are conducted for CVaR.

Strengths: The theoretical results are the main contribution of the paper, particularly Lemma 2 and Theorem 6. These are mostly straightforward Rademacher complexity analyses, from a quick glance at the proofs. The second contribution is the inverted OCE, which may be of relevance in machine learning problems. Since adjustments to the usual risk (expected loss) framework are currently of interest, this provides another useful perspective.

Weaknesses: I have a handful of minor concerns. (1) It would have been nice for the experiments to explore more than CVaR, since there are a number of OCEs that are given as examples. Exploring inverted OCEs would have been interesting too... (2) ... because while the OCE formulation is convex, at least in the loss, for CVaR (and probably entropic risk), the inverted OCEs look like they lead to a non-convex problem. While machine learning has learned to live with non-convexity in the models, some basic experiments could help assuage any concerns. (3) In many of the discussions of the generalization bounds, it seems like the paper would like to walk a non-existant tightrope between the empirical losses (and loss variances) and the Rademacher complexities. When using complicated neural networks, my understanding is that these bounds are mostly vacuous because the Rademacher complexities are high, hence the battles over rethinking generalization or the shortcomings of uniform convergence. I don't view these issues as meaning that we shouldn't examine these types of theory problems, but I find the suggestion that the empirical terms will simply vanish and this will solve all our problems to be disingenuous. Indeed, if a function class is a universal approximator, then its Rademacher complexity will likely be very large (assuming a reasonable data distribution). (4) The technical results are solid but don't seem to be particularly involved. This is fine, but it means that the results themselves have to be useful, which they may be.

Correctness: The paper looks essentially correct, although I did not read all the proofs in detail.

Clarity: The paper is fairly clearly written. It is definitely an above-average submission in this respect.

Relation to Prior Work: To the best of my knowledge, yes.

Reproducibility: Yes

Additional Feedback: I have read the other reviews and the response. I doubt the paper will be revolutionary (I've read maybe one such paper among the 30+ I've ever reviewed), but it's solid. Also, the authors put forth a good effort in their response (perhaps with a little too much ***-kissing though).


Review 4

Summary and Contributions: The paper discusses learning and generalization when the mean loss objective is replaced by a risk-sensitive objective with different weights attributed to data depending on the loss. Such occurs for example in various approaches to robust learning, where only a fraction of the sample with smallest losses is considered. The paper considers statistical functionals called optimized uncertainty equivalents (OCE) or inverse OCE's. These have a variational definition, and their minimization over a function class by minimizing their plug-in estimators is discussed. Excess risk bounds are given by way of uniform bounds depending on the Rademacher complexity of the function class. Learning guarantees for the usual average loss are also given for the OCE-minimizers, depending on the variance of the average-loss-minimizer (OCE) or on the empirical variance of the empirical OCE-minimizer (inverse OCE's). The paper suggests a connection to Sample-Variance-Penalization (SVP) and concludes with some experimental results. The appendix also contains an analysis of rubustness of some OCE-functionals in terms of influence functions.

Strengths: The OCE-functionals are well motivated and explained. I was unfamiliar with these concepts (originating in finance and economics) and learned quite a bit from the paper. The theoretical analysis is elegant and clear. The sound and convincing proofs in the appendix are a pleasure to read.

Weaknesses: I can see only very minor weaknesses. The notation could be explained a bit better, such as Lip(\phi) or the notation for quantiles. The equivalence (3) and (4) could be sketched. It would be much nicer to have a self-contained proof of Proposition 1. Also the connection to L-statistics could be mentioned in this context. ---------------------------update----------------------------------------------- I disagree with review #2. To me the reduction of the nonlinear objective to a linear one is simple and elegant. There is a wealth of methods to bound Rademacher averages, which can be combined with the results in the paper in a simple and efficient way. I wish to keep my score.

Correctness: Everything seems correct to me, but I didn't try to reproduce the numerical simulations.

Clarity: To me it seems as clear as possible, given the page limit and the material covered.

Relation to Prior Work: There is no section on "related work", but the discussion in the first paragraphs of Section 3 appears sufficient to me.

Reproducibility: Yes

Additional Feedback: To proceed from (33) it is not necessary to invoke Massart's lemma, but you can work directly with \cal{G}, using the triangle inequality and observing that E sup_\lambda \sum_i \epsilon_i \lambda \le M E |\sum_i \epsilon_i| \le M sqrt(n). This replaces the sqrt{8 log 2} by 2. I believe that in (36) log(2/\delta) should be log(1/\delta). The log(2/\delta) comes in after the union bound. In (52) you need a bar above the oce.

[Author Response · NeurIPS 2020]

We deeply appreciate constructive and insightful comments from the reviewers.

─────────────────────────────────────── **Reviewer #1** ───────────────────────────────────────

**Computing OCE minimizers.** On the one hand, we clarify that Theorems 3 and 6 apply to any approximate OCE minimizer, including those given by [40], and Lemma 2 applies to an arbitrary algorithm, even including those not minimizing OCE risks. Our results provide generalization guarantees with the same or tighter dependency for algorithms given by [40]. We will clarify this point in the revised version. On the other hand, we agree to the reviewer's point that clearly stating the optimization-relevant properties of (inverted) OCE risks may facilitate future works in this direction. To this end, we will survey existing results (in the appendix) that are relevant to the optimization of standard risk-sensitive measures, e.g., smoothness/convexity properties of CVaR, entropic risk, and mean-variance.

**Prior work on stability & convergence.** We believe that R1 refers to (Line 42), where we point to [30]. The work [30] considers an MDP setting that where the standard Bellman's optimality principle cannot handle risk-sensitive cases. By contrast, we focus on nonsequential scenarios where such concerns do not arise. We will clarify this point in a separated "Related Works" section, along with detailed discussions on relevant literature.

─────────────────────────────────────── **Reviewer #2** ───────────────────────────────────────

**Apparent lack of tailored analysis for risk-sensitivity.** We begin by noting that the risk-sensitivity is indeed distilled into the smoothness characteristics of the disutility $\phi(\cdot)$ in our results, instead of completely disappearing from the analysis. The proposed OCE framework enables a deceptively simple treatment of risk-sensitivity by relating the problem to risk-neutral learning via contraction principles. Such strategy (similar to what [D+19] did for adversarially robust learning) allows us to fully utilize accumulated insights on risk-neutral learning to understand risk-sensitive learning, without having to resort to overly complicated machineries (e.g., adapting Dvoretzky-Kiefer-Wolfowitz inequality to consider data-dependent weights). Given the significance of the considered problem, we view this as an *advantage* brought by our framework, instead of a weakness. Nevertheless, we also provide several "tailored" proof tools, including *product hypothesis space analysis* (Lines 430–438) and *two-sided variance-based characterization of OCE* (Appendix A.5), may be of readers' technical interest. We will highlight these points clearly in the main text.

**Vacuity of Rademacher-style analysis for DNNs, and realizability.** We thank R2 for pointing this out. While recent progress [S+20, N+20] shed new light on the power of Rademacher-style uniform convergence analysis for DNNs, we agree to the point that the current tone of the manuscript may be too bold. We will revise the manuscript to avoid exemplifying DNNs to make sense of realizability conditions.

**Related works and connection to fair ML.** Thank you for this valuable suggestion. We will add a standalone "Related Works" section to provide in-depth comparisons with existing statistical learning literature. Also, we will establish explicit connections to the fairness risk measures axiomatically defined in [49].

─────────────────────────────────────── **Reviewer #3** ───────────────────────────────────────

**Experiments other than CVaR (points 1&2).** Although our main scope is on a theoretical side than an algorithmic side, we agree to the reviewer's point that clarifying optimization properties of OCE risks may help readers grasp the potentials of our framework. For this purpose, we will make the following two revisions: (1) We will give a pointer to the concurrent work of Li et al. [L+20] (which appeared on arXiv after the submission deadline). Via large-scale experiments [L+20], the authors empirically observe that *both entropic risk and its inverse* can be optimized efficiently via standard mini-batch gradient descent, as R3 correctly anticipated. (2) For completeness, we will additionally survey existing theoretical results (in the appendix) that are relevant to the optimization for standard risk-sensitive measures.

**Vacuity of Rademacher complexities.** We resonate with your concern that any argument regarding the generalization properties of deep neural networks requires a delicate care. We will revise the manuscript to avoid exemplifying neural networks to support the applicability of realizability conditions.

**Technical benefits.** As R3 keenly points out, our primary focus is to establish an effective theoretical framework to formalize risk-sensitive learning, instead of pursuing technical exquisiteness. Nevertheless, we believe that several proof techniques may be of theoreticians' interest; for details, see Lines 22–24 of this response.

─────────────────────────────────────── **Reviewer #4** ───────────────────────────────────────

**Suggestions on discussions and proofs.** We express our deepest gratitude for the detailed comments, especially on the ideas to refine our result and the pointer to L-statistics/estimators. All these valuable comments will be incorporated.

─────────────────────────────────────── **Additional References** ───────────────────────────────────────

[D+19] Y. Dong et al. Rademacher complexity for adversarially robust generalization. In *ICML*, 2019.

[L+20] T. Li et al. Tilted empirical risk minimization. arXiv preprint 2007.01162, 2020.

[N+20] J. Negrea et al. In defense of uniform convergence: Generalization via derandomization with an application to interpolating predictors. In *ICML*, 2020.

[S+20] T. Suzuki et al. Compression based bound for non-compressed network: unified generalization error analysis of large compressible deep neural network. In *ICLR*, 2020.


[Meta-Review · NeurIPS 2020]

This is a learning theory paper in situation where the usual mean loss objective function is replaced by a risk-sensitive objective with different weights attributed to data depending on the loss. This setting is of high importance in robust learning, where only a fraction of the sample with smallest losses is considered. This paper provides an analysis of this setting via Rademacher bounds. The paper suggests a connection to Sample-Variance-Penalization (SVP) and concludes with some experimental results. The appendix also contains robustness analysis. Robust learning is one of the few new issues that rise in importance for our community, and I think this work shed some new interesting lightning about it. Only one reviewer score slightly under acceptation (with a 5), he nevertheless agree for poster acceptance in the post-rebuttal discussion. The main weakness he proposed was: β€œI believe that the main weakness of the paper is that on a technical level, the results (lemma 2 and theorem 3) are just direct extensions of classic uniform convergence arguments based on Rademacher complexity.” I, and at least one other reviewers think that this should not be consider as a weakness: β€œTo me the reduction of the nonlinear objective to a linear one is simple and elegant. There is a wealth of methods to bound Rademacher averages, which can be combined with the results in the paper in a simple and efficient way.” Hence, and especially because there is a need in our community for a better understanding of robust learning and of out-of-distribution learning, I recommend poster acceptance.